# Exploring patient characteristics and respiratory impacts of pulmonary melioidosis: A 5-year experience from endemic region of Thailand

Jinjuta Ngeyvijit[1], Chawakorn Payackpunth[2], Pattawee Saengmongkonpipat[2], Subencha Pinsai[2], Antenor Rodrigues[3,4], Vorakamol Phoophiboon[3,4,5]*

1 Division of Pulmonary and Critical Care Medicine, Department of Medicine, Chaophraya Abhaibubejhr, Prachinburi, Thailand, 2 Department of Medicine, Chaophraya Abhaibubejhr, Prachinburi, Thailand, 3 Keenan Centre for Biomedical Research, Li Ka Shing Knowledge Institute, Unity Health Toronto, Toronto, Ontario, Canada, 4 Interdepartmental Division of Critical Care Medicine, University of Toronto, Toronto, Ontario, Canada, 5 Division of Critical Care Medicine, Department of Medicine, Faculty of Medicine, Chulalongkorn University, Thailand

* vorakamol.phoophiboon@unityhealth.to

## Abstract

### Background and objective

Pulmonary melioidosis, caused by *Burkholderia pseudomallei*, emerges as the most prevalent and highly fatal manifestation of melioidosis. Despite being a global health concern, pulmonary melioidosis remains insufficiently studied and reported.

### Research question

Whether patients with pulmonary melioidosis have any potential characteristics or specific risk factors that are associated with their respiratory outcome's trajectory.

### Methods

A 5-year retrospective cohort study was conducted. Electronic medical records of patients admitted for *Burkholderia pseudomallei* pneumonia between January 1, 2019 and December 31, 2023, at Chaophraya Abhaibhubejhr hospital, Prachinburi, Thailand were reviewed.

### Results

Of 1,486 adults admitted with bacterial pneumonia, 36 patients (2.4%) were diagnosed with microbiologically confirmed pulmonary melioidosis. Thirty of 36 patients (83.3%) developed acute respiratory failure requiring mechanical ventilation within 24 hours after admission. Twenty-three of 36 (63.9%) met the 2024 global definition of acute respiratory distress syndrome (ARDS) with 78% being moderate-to-severe ARDS. Pulmonary melioidosis with ARDS was associated with excessive alcohol consumption ($p = 0.013$) and body mass index (BMI) < 20 Kg/m$^2$ (odds ratio of 6,

which permits unrestricted use, distribution, and reproduction in any medium, provided the original author and source are credited.

**Data availability statement:** All data are in the manuscript and/or supporting information files.

**Funding:** AR is supported by a Canadian Institutes of Health Research (CIHR) Fellowship, Canada (#187900). The funder had no role in study design, data collection and analysis, decision to publish or preparation of the manuscript.

**Competing interests:** The authors have declared that no competing interests exist.

95%CI 1.080-33.321, $p = 0.041$) when compared to non-ARDS. BMI < 20 Kg/m$^2$ was also associated with ARDS development, independent of age (odds ratio of 29.27 (95%CI 1.849 – 463.678, $p = 0.017$). The overall mortality rate of pulmonary melioidosis was 55.6%, with no differences between patients with ARDS and non-ARDS (65.2% vs 38.5%, respectively; $p = 0.169$).

## Conclusion

ARDS frequently emerges in patients with pulmonary melioidosis requiring invasive mechanical ventilation and intensive care support. Excessive alcohol consumption and a low BMI status were associated with the development of ARDS in these patients.

## Author summary

Melioidosis is a tropical infectious disease caused by *Burkholderia pseudomallei*. The diagnosis is confirmed by the growth of *B. pseudomallei*. Pulmonary melioidosis is the most common organ involvement of melioidosis with a high fatal rate. Acute respiratory distress syndrome (ARDS) is life-threatening lung injury characterized by poor oxygenation and non-compliant (stiff) lungs. Poor outcome has been reported in pulmonary melioidosis with ARDS. We aimed to identify clinical characteristics and potential risk factors between patients with pulmonary melioidosis with and without ARDS. Our study demonstrated that excessive alcohol consumption and a low BMI status in patients with pulmonary melioidosis could predict the development of ARDS.

## Introduction

Melioidosis is a bacterial infection caused by *Burkholderia pseudomallei.* A common route of infection occurs through a direct exposure (e.g., inhalation, non-intact skin, ingestion) with contaminated soil and surface water [1,2]. Melioidosis is known as "the great mimicker" of other infectious diseases due to non-specific clinical manifestations, making a challenge of diagnosis [2,3]. The difficulty also extends to a decision of management such as choice of antibiotics, duration of treatment, and assessment to clinical response [4]. Melioidosis is considered highly endemic to tropical regions, particularly in Southeast Asia and northern Australia [5]. A large Thai cohort study conducted over a 10-year period revealed a high incidence of melioidosis in the northeastern [6] and eastern regions [7], recognized as a hyperendemic zone, with high mortality rate of 25–40% [6,7]. The disease demonstrates increased prevalence during the rainy season. Given *B. pseudomallei* resides in soil and water, the increasing number of international travels, migrations, and importations of goods from endemic regions contributes to its widespread dissemination into non-endemic areas [8,9].

Pulmonary involvement is commonly reported in melioidosis with a mortality rate exceeding 40% [5,10–12]. Positive culture (from, i.e., sputum, bronchoalveolar lavage (BAL) fluid, or alveolar tissue) of *B. pseudomallei* is the gold standard for a diagnosis of pulmonary melioidosis. A serological test - indirect hemagglutination assay or melioidosis titer lacks both sensitivity and specificity of diagnosis, particularly in the endemic regions where individuals have been exposed to the organism without exhibiting the disease [13]. The number of known melioidosis's risk factors include diabetes mellitus, excessive alcohol consumption, chronic lung disease, malignancy and systemic corticosteroid use [2,14,15]. However, none of these factors have been thoroughly addressed in pulmonary melioidosis. Over the past 20 years, there have been a few cohort studies including case reports exclusively focused on clinical characteristics, risk factors and respiratory outcomes in pulmonary melioidosis. In addition, none have assessed the relationship between pulmonary melioidosis and acute respiratory distress syndrome (ARDS) [10,12,16–18]. Notwithstanding, an outcome's trajectory remains unpredictable as its incidence and number of studies are relatively low incidence and limited number of studies [2,6]. Pulmonary melioidosis remains understudied with scarce data currently available.

We conducted a 5-year retrospective study on patients with pulmonary melioidosis to assess their respiratory consequences including a burden of disease from the endemic region of Thailand. This study aimed to establish more comprehensive data of this globally emerging yet understudied and underreported disease.

## Materials and methods

### Ethics statement

The study was approved and granted a waiver of written informed consent by ethics committee of Chaophraya Abhaibhubejhr hospital (IRB-BHUBEJHR-322), Prachinburi, Thailand. As this retrospective observational study involved less than minimal risk to participants, which was conducted using de-identified data (not including any live participants) and adhered to the principles of Declaration of Helsinki [19].

### Study design and subjects

We conducted a retrospective cohort study of adults with bacterial pneumonia who required hospitalization, between January 1, 2019 and December 31, 2023, at Chaophraya Abhaibhubejhr hospital, Prachinburi, Thailand. This tertiary care center is considered as one of the largest endemic areas of Thailand for melioidosis. Patients were included if they were at least 18 years of age with positive sputum or BAL fluid culture for *B. pseudomallei* and presented abnormal chest X-ray (CXR) findings associated with pneumonia. Patients were excluded if they were known to have palliative or end-of-life care. Electronic medical records (EMRs) were used to identify the disease diagnosed with the International Classification of Disease, tenth revision (ICD-10) [20].

### Data collection

Demographic data (age, sex, weight, height, body mass index (BMI) and comorbidities), history of excessive alcohol consumption, history of current smoking, Sequential (sepsis-related) Organ Failure Assessment (SOFA) score, biochemical and microbiologic variables, chest radiograph findings, hospital length of stay (LOS), intensive care unit (ICU) LOS, 48-hour use of vasopressor, treatment during hospitalization and 28-day ventilator free days (VFDs) and mortality were collected from the EMRs. Excessive alcohol consumption was defined as binge drinking (≥ 4 standard drinks for women or ≥ 5 standard drinks for men during an occasion) or heavy drinking (≥ 8 standard drinks for women or ≥ 15 standard drinks for men during a week) [21,22]. Current smoking was self-reported as "yes" or "no" question. Gold standard of bacterial detection was identified by bacteria culture using Matrix-Assisted Laser Desorption Ionization time of Flight Mass Spectrometry (MALDI-TOV MS) technique [23]. MALDI-TOV-MS is a rapid bacterial identification technique from cultured colonies, it offers faster turnaround time and more accurate identification compared to conventional method. This technique is commonly used in clinical microbiology laboratories.

We included patient's respiratory variables: mode and level of oxygen therapy, arterial blood gas, ratio of oxygen saturation to fraction of inspired oxygen ($SpO_2/FiO_2$ or SF ratio), ratio of partial pressure of oxygen to fraction of inspired oxygen ($PaO_2/FiO_2$ or PF ratio) and intubation outcome. The definition of ARDS in our study followed by a new global definition of ARDS [24]: i) include high flow nasal cannula with a minimum of flow rate of ≥ 30 L/min, ii) identify hypoxemia by SF ratio ≤ 315 (if oxygen saturation as measured by pulse oximetry is ≤ 97%) and/or PF ratio ≤ 300 mm Hg, iii) retain bilateral opacities of lung but add lung ultrasound as imaging modality, and iv) expect acute or worsening of hypoxemic respiratory failure within 1 week. The diagnosis of ARDS was determined by two independent investigators (JN and PS). In the event of discordance diagnosis, a third independent investigator as a pulmonologist (VP) was engaged to reassess the radiographic findings and ascertain the final diagnosis.

## Statistical analysis

Statistical analysis was performed with STATA 18.0 (StataCorp, College Station, TX, USA). Descriptive statistics were employed to summarize demographic data and baseline characteristics. Continuous variables were expressed as mean and standard deviation or median and interquartile range, as appropriate. Categorical variables were expressed as absolute numbers and percentages. Normal distribution of data was verified by Shapiro-Wilk test. Differences between ARDS and non-ARDS groups among pulmonary melioidosis patients were evaluated to identify potential predictors of ARDS. For continuous variables, the t-test was used for parametric data and the Mann-Whitney U test for non-parametric data. Categorical variables were analysed using the Chi-square test or Fisher's exact test, as appropriate. Univariate logistic regression analysis was conducted to examine the potential independent factors associated with risk of ARDS development. We dichotomized patients according to BMI < 20 Kg/m² (derived from median BMI of ARDS patients), SOFA score > 11 [25], lactate level > 4 mmol/L [26]. Age and platelet count were calculated as continuous data. A multivariate logistic regression was also performed. Based on rule of thumb for 10 events per one variable [27], 2 potential factors: age per 5-year increase and BMI < 20 Kg/m² were included. Statistical significance was defined as a *p*-value of less than 0.05.

## Results

From January 1, 2019, to December 31, 2023, 1,486 patients admitted due to bacterial pneumonia were screened through the EMRs, 40 patients (2.7% of bacterial pneumonia) presented with positive sputum or BAL fluid for *B. pseudomallei*. Thirty-six (2.4%) patients were included for data analysis. Four patients were excluded: age < 18 years (2 patients), *B. pseudomallei* colonization (2 patients; they were diagnosed with aspiration pneumonitis given their clinical presentations and rapid improvement of symptoms and CXR, without anti-viral or bacterial treatment.) (Fig 1)

Baseline characteristics are described in Table 1. The median age was 59 [52–64] years, 26 (72.2%) were male. Diabetes mellitus (23, 63.9%), excessive alcohol consumption (8, 22.2%) and chronic kidney disease (3, 8.3%) were the common comorbidities. The duration of ICU and hospital LOS of the survival cases were 10.0 [3.5-17.5] and 14.0 [13.0-22.0] days, respectively.

We classified patients into 2 groups based on severity of respiratory presentation: non-ARDS and ARDS. Patients with ARDS had lower BMI compared to non-ARDS, 19.8 [17.0-24.8] vs 23.2 [20.9-27.6] Kg/m², *p* = 0.038. All patients with excessive alcohol consumption (n = 8) were reported in ARDS group, *p* = 0.013. The SOFA score at the first 24 hours of hospital presentation was higher in ARDS 8 [6–12] vs in non-ARDS 4 [3–8] points, *p* = 0.033. Initial vasopressor use and serum lactate level were not different between groups. Survival patients with ARDS required longer hospital LOS compared to non-ARDS patients, 23 [13–34] vs 14 [14–15] days, respectively (*p* = 0.001). The overall 28-day mortality rate was 44.4% with no differences between groups, 65.2% in ARDS vs 38.5% in non-ARDS, *p* = 0.169. When categorized patients with ARDS by severity (Fig 2, n = 23), the mortality rates were 8.7%, 13% and 43.5%, in mild, moderate and severe ARDS, respectively (*p* = 0.040).

PLOS Neglected Tropical Diseases

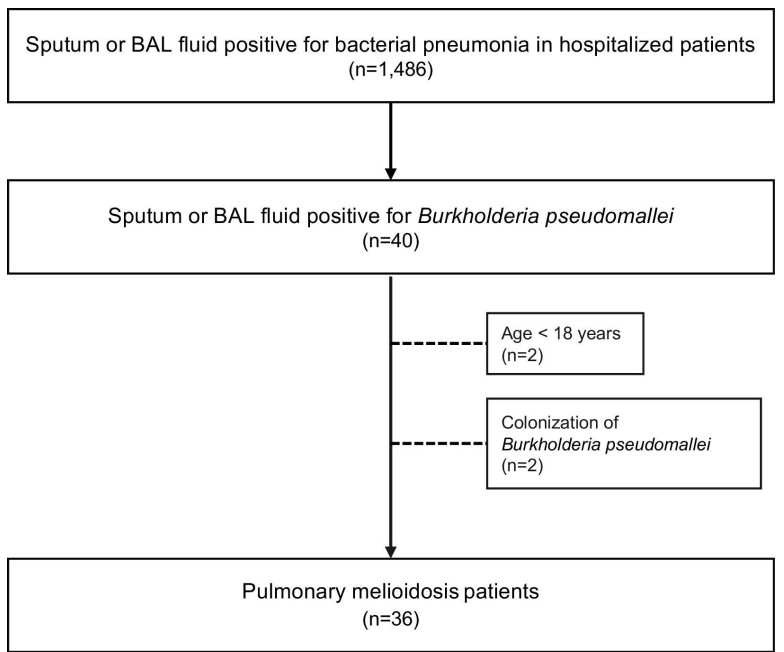

**Fig 1. Flow diagram of patients.** Adult patients with positive sputum or bronchoalveolar lavage fluid for pulmonary melioidosis were identified (n = 36) after excluding those younger than 18 years of age and those with colonization of *Burkholderia pseudomallei*.

Twenty-three patients developed ARDS according to the new 2024 global definition of ARDS (18 patients in the first 12 hours and 5 patients in 24 hours after admission). Four of 5 ARDS patients (80%) who were initially on high flow nasal cannula required escalation to mechanical ventilation at 12–24 hours after admission (Fig 3).

Respiratory, laboratory and microbiologic parameters are described in Table 2. A higher number of patients with ARDS (n = 22 of 23, 95.7%) required invasive mechanical ventilation support than patients with non-ARDS (n = 8 of 13, 61.5%), $p = 0.016$. There were no differences in PF or SF ratio at the first presentation between non-ARDS and ARDS. VFDs in non-ARDS were longer than ARDS group, 27 [25–28] vs 25 [18–27], respectively, $p = 0.014$. However, VFDs in survival patients were not different between groups ($p = 0.784$). The patients with ARDS reported more quadratic involvement (both upper and lower-lobe involvement) at the first CXR during the admission compared to patients with non-ARDS, $p = 0.008$. In non-ARDS patients, isolated upper-lobe involvement (53.8%) was more predominant than isolated lower-lobe involvement (23.1%).

Twenty-eight (77.8%) of pulmonary melioidosis patients demonstrated coexisting *B. pseudomallei* bacteremia. *Streptococcus viridans, Klebsiella pneumoniae,* and *Acinetobactor baumanii* were concomitantly detected in sputum cultures of patients with pulmonary melioidosis. To note, sputum samples were collected upon initial presentation at emergency room and none had predisposing risk factors of nosocomial infections; thus, all organisms identified in this study are from the community. Two ARDS patients were concomitantly infected with Coronavirus disease 19 (COVID-19), which standard anti-viral treatment was given. Ceftazidime and meropenem were the main intravenous antibiotic treatment during hospitalization (intensive phase). Meropenem was selected in patients who were concerned of worsening and/or severe disease (e.g., severe sepsis, worsening respiratory failure, high risk of drug resistance), 14 (87.5%) of 16 patients were prescribed to meropenem were in ARDS group, $p = 0.014$. Duration of intravenous antibiotic in intensive phase was 14 [14–14] days (Table 3).

The univariate logistic regression analysis showed that BMI was associated with ARDS development, odds ratio of 0.81 (95%CI 0.668 – 0.982, $p = 0.032$). BMI < 20 Kg/m$^2$ was independently associated with a greater risk of ARDS, odds ratio

**Table 1. Baseline characteristics.**

| | All (n = 36) | Non-ARDS (n = 13) | ARDS (n = 23) | p-value |
|---|---|---|---|---|
| Age, years | 59 [52-64] | 58 [52-64] | 60 [50-66] | 0.986 |
| Male, n (%) | 26 (72.2) | 10 (79.9) | 16 (70.0) | 0.636 |
| BMI, Kg/m$^2$ | 22.5 [18.6-25.0] | 23.2 [20.9-27.6] | 19.8 [17.0-24.8] | **0.038** |
| Current smoking, n (%) | 12 (33.3) | 3 (23.1) | 9 (39.1) | 0.468 |
| Excessive alcohol consumption, n (%) | 8 (22.2) | 0 | 8 (34.8) | **0.013** |
| Underlying disease, n (%) | | | | |
| Transfusion dependent | 0 | 0 | 0 | – |
| HIV positive | 0 | 0 | 0 | – |
| Diabetes mellitus | 23 (63.9) | 8 (61.5) | 15 (65.2) | 0.890 |
| Coronary vascular disease | 1 (2.8) | 0 | 1 (4.4) | 0.446 |
| Heart failure and cardiomyopathy | 1 (2.8) | 0 | 1 (4.4) | 0.446 |
| COPD | 1 (2.8) | 0 | 1 (4.4) | 0.446 |
| Asthma | 2 (5.6) | 0 | 2 (9.0) | 0.274 |
| ILD | 0 | 0 | 0 | – |
| Active lung cancer | 2 (5.6) | 0 | 2 (8.7) | 0.274 |
| ESRD with dialysis dependent | 0 | 0 | 0 | – |
| Chronic kidney disease | 3 (8.3) | 1 (7.7) | 2 (8.7) | 0.917 |
| SOFA score* | 7 [5-10] | 4 [3-8] | 8 [6-12] | **0.033** |
| Vasopressor used in 48 hours, n (%) | 20 (55.6) | 6 (46.2) | 14 (60.9) | 0.393 |
| Highest serum lactate level at the first 24 hours, mmol/L | 3.6 [2.2-7.6] | 2.5 [1.3-6.3] | 3.8 [3.2-8.5] | 0.097 |
| ICU LOS, days | n = 30; 3.0 [1.0- 9.8] | n = 8; 2.0 [1.0-8.8] | n = 22; 4.0 [1.0-11.5] | 0.375 |
| ICU LOS of survival cases, days | n = 10; 10.0 [3.5-17.5] | n = 7; 10.0 [1.0-10.0] | n = 3; 11.0 [3.0-29.0] | 0.405 |
| Hospital LOS, days | 13.0 [1.0-16.0] | 14.0 [3.0-14.0] | 10.0 [1.0-21.0] | 0.573 |
| Hospital LOS of survival cases, days | n = 16; 14.0 [13.0-22.0] | n = 8; 14.0 [14.0-15.0] | n = 8; 23.0 [13.0-34.0] | **0.001** |
| 28-days mortality, n (%) | 20 (55.6) | 5 (38.5) | 15 (65.2) | 0.169 |

Data are presented as mean (SD) or median [interquartile range] or n (%). Abbreviation: ARDS = acute respiratory distress syndrome, BMI = body mass index, COPD = chronic obstructive lung disease, ILD = interstitial lung disease, ESRD = end stage renal disease, ICU = intensive care unit, LOS = length of stay, SOFA = sequential organ failure assessment.

* Worst values at the first 24 hours of hospital presentation.

of 6 (95%CI 1.080-33.321, $p = 0.041$). SOFA score [25], lactate level [26], platelet count, male, diabetes mellitus, chronic kidney disease, bacterial respiratory co-infections and coexisting *B. pseudomallei* bacteremia were not associated with ARDS development in patients with pulmonary melioidosis. Multivariate logistic regression analysis revealed that BMI < 20 Kg/m² was associated with ARDS development, independent of age; odds ratio of 29.27 (95%CI 1.849 – 463.678, $p = 0.017$). (Table 4)

## Discussion

Global attentiveness of pulmonary melioidosis is increasing as it is the most common organ involvement in melioidosis (> 50%), with a high mortality rate [1,6,11]. Our study focused on patients with microbiologically confirmed pulmonary melioidosis, to identify patient's characteristics, clinical outcomes and respiratory impacts of these patients. The patients with low BMI status (particularly, when BMI < 20 Kg/m²) or with history of excessive alcohol consumption were potentially associated with ARDS development.

Several studies have found strong associations between BMI and all-cause mortality, including patients with respiratory diseases [28–31]. Most have described a U-shaped weight of 20–25 Kg/m² associated with a lower mortality rate. Not

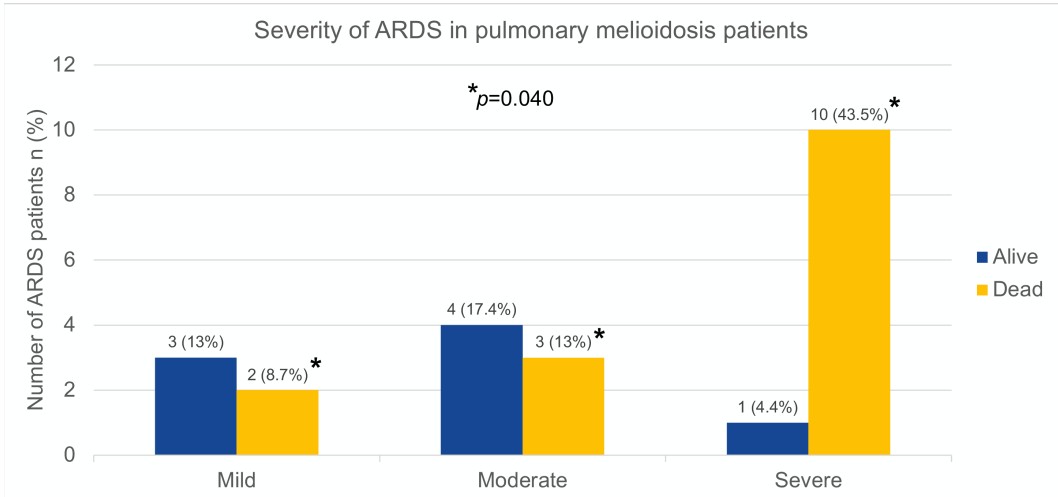

**Fig 2. Severity of ARDS in pulmonary melioidosis patients.** The dark blue bar represented the number of survival patients and the yellow bar represented the number of dead patients in a group of acute respiratory distress syndrome (ARDS). Comparing severities of ARDS (mild, moderate and severe) with chi-square test reported $p = 0.04$. Abbreviation: ARDS, acute respiratory distress syndrome.

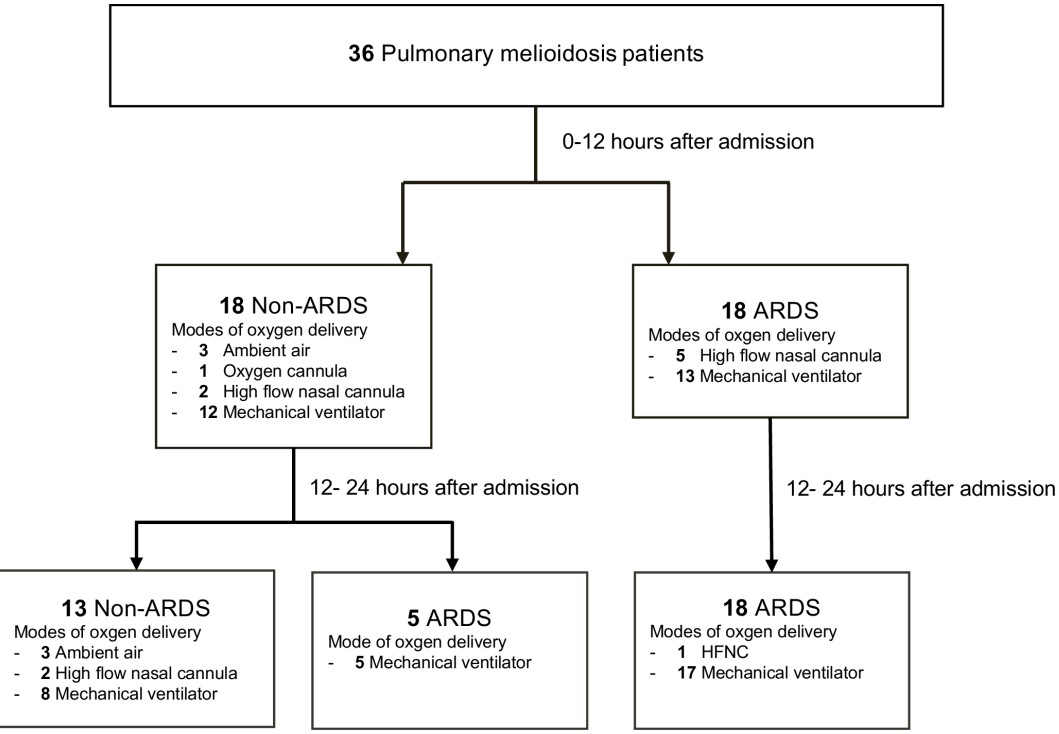

**Fig 3. Clinical course of pulmonary melioidosis.** Pulmonary melioidosis with respiratory clinical course of non-acute respiratory distress syndrome (ARDS) and ARDS patients at 0-12 and 12-24 hours after admission. The total number of patients with non-ARDS = 13 patients and with ARDS = 23 patients.

**Table 2. Respiratory and laboratory parameters.**

| | All (n = 36) | Non-ARDS (n = 13) | ARDS (n = 23) | p-value |
|---|---|---|---|---|
| **Respiratory variables** | | | | |
| PaO$_2$/FiO$_2$ ratio at first presentation | n = 26 | n = 8 | n = 18 | 0.213 |
| | 132 | 211 | 124 | |
| | [96-241] | [117-316] | [94-195] | |
| PaO$_2$/FiO$_2$ ratio, worst values | n = 30 | n = 8 | n = 22 | 0.237 |
| | 111 | 131 | 106 | |
| | [59-182] | [74-324] | [54-164] | |
| SaO$_2$/FiO$_2$ ratio at first presentation | 181 | 240 | 164 | 0.090 |
| | [136-145] | [156-248] | [98-238] | |
| SaO$_2$/FiO$_2$ ratio, worst values | 139 | 221 | 97 | **0.022** |
| | [86-241] | [106-380] | [79-232] | |
| Mechanical ventilator, n (%) | 30 (83.3) | 8 (61.5) | 22 (95.7) | **0.016** |
| Ventilator-free day, days | 26 [21-27] | 27 [25-28] | 25 [18-27] | **0.014** |
| Ventilator-free day in survival case, days | n = 16; 25 [21–28] | n = 8; 28 [25–28] | n = 8; 20 [16–25] | 0.784 |
| **First chest X-ray findings during admission** | | | | |
| Predominance, n (%) | | | | **0.008** |
| Upper lung | 12 (33.3) | 7 (53.8) | 5 (21.7) | |
| Lower lung | 6 (16.7) | 3 (23.1) | 3 (13.0) | |
| Both upper and lower lungs | 16 (44.4) | 1 (7.7) | 15 (65.2) | |
| No infiltrations | 2 (5.6) | 2 (15.4) | 0 | |
| Distribution, n (%) | | | | **<0.001** |
| Unilateral | 13 (36.1) | 10 (76.9) | 3 (13.0) | |
| Bilateral | 21 (58.3) | 1 (7.7) | 20 (87.0) | |
| **Hematological and biochemical findings** | | | | |
| White blood cell count, cells/mm$^3$ | 14,750 [7,560- 19,513] | 14,310 [9,730-19,210] | 15,190 [5,940-19,850] | 0.974 |
| Hemoglobin, g/dL | 11.6 [9.8-13.1] | 11.4 [10.3-12.0] | 11.9 [9.2-13.9] | 0.668 |
| Platelet, cells/mm$^3$ | 202,500 [99,750-260,000] | 245,000 [199,000-361,000] | 169,000 [79,000-236,000] | 0.093 |
| Total bilirubin, g/dL | 0.7 [0.4-1.5] | 0.7 [0.5-1.1] | 0.9 [0.3-2.4] | 0.403 |
| SGOT, U/L | 82 [32.5-175] | 51 [28-197] | 88 [33-173] | 0.578 |
| SGPT, U/L | 47 [21-78] | 45 [19-89] | 47 [20-76] | 0.958 |

Data are presented as mean (SD) or median [interquartile range] or n (%). Abbreviation: ARDS = acute respiratory distress syndrome, MV = mechanical ventilation, SGOT = serum glutamic-oxaloacetic transaminase, SGPT = serum glutamate-pyruvate transaminase.

only obesity is addressed as a risk factor of ARDS [32–34], underweight status is independently associated with decreasing pulmonary function [35] and increasing risk of ARDS, particularly in the elderly [36]. Our findings are consistent with previously published data on bacterial pneumonia, showing that BMI < 20 Kg/m$^2$ was independently associated with ARDS development in pulmonary melioidosis patients.

Diabetes mellitus, excessive alcohol consumption and chronic kidney disease were known as risk factors of melioidosis [2,37]; however, only excessive alcohol consumption was associated with ARDS development in our study (melioidosis with pulmonary involvement). In animal model, binge alcohol consumption contributed to an increase in *B. pseudomallei* colonization in lung tissue [38]. As a result, it may be challenging in some cases with history of excessive alcohol consumption (high risk of aspiration) to distinguish between colonization of *B. pseudomallei* from the chronic aspiration vs pulmonary melioidosis [39].

**Table 3. Microbiologic findings and pulmonary melioidosis treatment.**

| | All (n = 36) | Non-ARDS (n = 13) | ARDS (n = 23) | p-value |
|---|---|---|---|---|
| Microbiologic findings | | | | |
| *Burkholderia pseudomallei* bacteremia, n (%) | 28 (77.8) | 10 (76.9) | 18 (78.3) | 0.926 |
| Concomitant positive sputum cultures, n (%) | 13 (56.5) | 6 (46.2) | 7 (30.4) | 0.346 |
| *Streptococcus viridans* | 5 (38.5) | 3 (50.0) | 2 (28.6) | |
| *Klebsiella pneumoniae* | 3 (23.1) | 1 (16.7) | 2 (28.6) | |
| *Acinetobactor baumanii* | 3 (23.1) | 2 (33.3) | 1 (14.3) | |
| *Escherichia coli* | 1 (7.7) | 1 (16.7) | 0 | |
| *Morganella morganii* | 1 (7.7) | 0 | 1 (14.3) | |
| Positive COVID-19 test, n (%) | 2 (5.6) | 0 | 2 (8.7) | 0.525 |
| Positive from nasopharyngeal swab, RT-PCR | 1 (2.8) | 0 | 1 (4.3) | |
| Positive from nasopharyngeal swab, ATK | 1 (2.8) | 0 | 1 (4.3) | |
| Treatment during hospitalization (intravenous intensive phase) | | | | |
| Antibiotics, n (%) | | | | |
| Ceftazidime | 20 (66.7) | 9 (69.2) | 9 (39.1) | 0.164 |
| Meropenem* | 16 (44.4) | 2 (15.4) | 14 (60.9) | **0.014** |
| Duration of antibiotic, days | 13 [1-14] | 14 [13-14] | 10 [1-14] | 0.293 |
| Duration of antibiotic in survival cases, days | n = 16; 14 [14–14] | n = 8; 14 [14–14] | n = 8; 14 [11–27] | 0.980 |

Data are presented as mean (SD) or median [interquartile range] or n (%). Abbreviation: ARDS = acute respiratory distress syndrome, COVID-19 = coronavirus disease 2019, RT-PCR = real-time polymerase chain, reaction, ATK = rapid antigen test. *Meropenem was used or stepped-up in worsening and/or severe disease (e.g., severe sepsis, worsening respiratory failure, high risk of drug resistance).

The SOFA score is one of the sepsis-related prediction models based on scores of multiple organ dysfunction including respiratory system deterioration which could predict ICU morbidity and mortality, particularly the score derived from the first few days [25,40]. In our study, patients with ARDS had higher SOFA score compared to non-ARDS group. However, in our univariate model, the SOFA score was not identified as an independent factor of ARDS development.

The Darwin melioidosis guideline has recommended two phases of melioidosis treatment: 1. intravenous intensive phase (a minimum of 2-week duration) and 2. subsequent oral eradication phase (at least 12 weeks) [41]. At intensive phase, ceftazidime is an initial drug of choice in most cases whereas meropenem (theoretically, more active to decrease endotoxin release) is reserved for those with severe disease or high risk of drug resistance [4,5,42]. Since a treatment regimen requires longer duration of intravenous route compared to other bacterial pneumonia [5], the hospital LOS of survival patients in these patients will be at least 2 weeks.

In our study, survival cases of ARDS in pulmonary melioidosis had a longer hospital LOS (23 [13–34] days) than previously published data ARDS cases in community acquired pneumonia (CAP); 16 [9–30] days) [43]. However, survival patients with non-ARDS in pulmonary melioidosis had similar hospital LOS to non-ARDS cases in CAP, 14 days approximately [43].

The number of hospitalized patients with pulmonary melioidosis in our cohort, requiring ICU admission and invasive mechanical ventilation support was higher than hospitalized patients with CAP, 83% vs 33%, respectively [43]. Particularly, 96% of ARDS patients with pulmonary melioidosis required this invasive mechanical ventilation support while a smaller number (29%) was demonstrated on ARDS patients with CAP. These findings demonstrated a significant burden of disease compared to other types of bacterial pneumonia.

In our study, concomitant community-acquired pathogens in pulmonary infection, defined as organisms isolated from specimens collected within 48 hours of hospital admission [44], such as *Streptococcus viridans*, *Klebsiella pneumoniae*, and *Acinetobacter baumannii*, were also identified (56.5%). Although these organisms are generally susceptible to

**Table 4. Factors associated with ARDS development in pulmonary melioidosis patients.**

| Univariate analysis | Odds Ratio | 95% Confidence Interval | p-value |
|---|---|---|---|
| Age (years) | 1.00 | 0.947 – 1.005 | 1.055 |
| Male | 0.69 | 0.143 – 3.284 | 3.284 |
| BMI (Kg/m²) | | | |
| ● As a continuous variable | 0.81 | 0.668 – 0.982 | **0.032** |
| ● *BMI < 20 Kg/m² | 6.00 | 1.080 – 33.321 | **0.041** |
| SOFA score | | | |
| ● As a continuous variable | 1.37 | 0.948 – 1.989 | 0.094 |
| ● SOFA score > 11 points | 2.31 | 0.219 – 24.316 | 0.486 |
| Lactate level (mmol/L) | | | |
| ● As a continuous variable | 1.11 | 0.921 – 1.328 | 0.280 |
| ● Lactate > 4 mmol/L | 1.33 | 0.337 – 5.273 | 0.682 |
| Excessive alcohol consumption** | – | – | – |
| Diabetes mellitus | 1.13 | 0.212 – 5.969 | 0.890 |
| Chronic kidney disease | 1.14 | 0.094 – 13.965 | 0.907 |
| Platelet count (cells/mm³) | 1.00 | 0.999 - 1.000 | 0.157 |
| Bacterial respiratory co-infections | 0.51 | 0.125 – 2.083 | 0.349 |
| Coexisting *Burkholderia pseudomallei* bacteremia | 1.08 | 0.212 – 5.494 | 0.926 |
| **Multivariate analysis** | | | |
| Age, per 5-year increase | 0.90 | 0.609 – 1.346 | 0.621 |
| BMI < 20 Kg/m² | 29.27 | 1.849 – 463.678 | **0.017** |

Data is shown as an estimated odds ratio with 95% confidence intervals of explanatory variables in the association with acute respiratory distress syndrome (ARDS) development of pulmonary melioidosis. *BMI < 20 Kg/m² was derived from a median BMI of ARDS patients (19.8 Kg/m²). **All excessive alcohol consumption patients developed ARDS. Age and platelet count was calculated as continuous data. Abbreviation: BMI = body mass index, SOFA = sequential organ failure assessment.

antibiotics commonly used in the treatment of melioidosis, such as ceftazidime and meropenem, the presence of multiple pathogens may contribute to more severe clinical presentations compared to solo pathogen. In addition, local infection control regulations, particularly for hospital acquired pneumonia and ventilator associated pneumonia (VAP), are actively implemented. These include adherence to hand hygiene protocol and VAP care bundle [44]. Further research into the impact of coinfections may offer valuable insights to enhance clinical management and guide therapeutic decisions in patients with melioidosis.

Although our study provides important and promising data on patients with pulmonary melioidosis, it has a number of limitations. First, the study was based on retrospective data collected from a single center, which may limit the generalizability of the findings to broader populations. Second, the relatively small number of patients may have reduced statistical power, potentially obscuring significant factors related to pulmonary melioidosis. However, our center is one of the biggest centers located in the regions where the prevalence of the disease is the highest. Third, although a low BMI was associated with ARDS development, independent of age, the interpretation of this multivariate analysis should be approached with caution due to the small number of patients and the wide range of 95% confidence interval. Our study also has several strengths. First, we used a new definition of ARDS to eliminate major limitations of the Berlin criteria (i.e., resource-limited settings, use of non-invasive support at the time of diagnosis). Second, this is the first study to identify patient's characteristics, risk factors associated with pulmonary melioidosis, and clinical and respiratory outcomes in a wide spectrum of diseases (uncomplicated pneumonia to ARDS). However, further research on a larger scale, involving longer periods of data collection and collaborating across multiple centers from the endemic regions is warranted.

## Conclusion

This is the first study assessing the clinical characteristics and respiratory outcomes of pulmonary melioidosis. Pulmonary melioidosis explicitly exhibited a high burden of disease leading to acute respiratory failure and mortality. Lower BMI status and excessive alcoholic consumption were associated with ARDS development.

## Supporting information

**S1 Data. Respiratory impacts of pulmonary melioidosis.**
(XLSX)

## Author contributions

**Conceptualization:** Jinjuta Ngeyvijit, Vorakamol Phoophiboon.

**Data curation:** Jinjuta Ngeyvijit, Chawakorn Payackpunth, Pattawee Saengmongkonpipat.

**Formal analysis:** Subencha Pinsai, Antenor Rodrigues, Vorakamol Phoophiboon.

**Investigation:** Jinjuta Ngeyvijit, Chawakorn Payackpunth, Pattawee Saengmongkonpipat, Subencha Pinsai, Vorakamol Phoophiboon.

**Methodology:** Jinjuta Ngeyvijit, Vorakamol Phoophiboon.

**Project administration:** Jinjuta Ngeyvijit, Vorakamol Phoophiboon.

**Supervision:** Vorakamol Phoophiboon.

**Validation:** Jinjuta Ngeyvijit, Vorakamol Phoophiboon.

**Visualization:** Vorakamol Phoophiboon.

**Writing – original draft:** Jinjuta Ngeyvijit, Antenor Rodrigues, Vorakamol Phoophiboon.

**Writing – review & editing:** Jinjuta Ngeyvijit, Vorakamol Phoophiboon.

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
