## [Decision Letter · Decision Letter 0]

PNTD-D-24-01573

Exploring Patient Characteristics and Respiratory Impacts of Pulmonary Melioidosis: A 5-Year Experience from Endemic Region of Thailand

Dear Dr. Phoophiboon,

Thank you for submitting your manuscript to PLOS Neglected Tropical Diseases. After careful consideration, we feel that it has merit but does not fully meet PLOS Neglected Tropical Diseases's publication criteria as it currently stands. Therefore, we invite you to submit a revised version of the manuscript that addresses the points raised during the review process.

Please submit your revised manuscript within 60 days May 24 2025 11:59PM. If you will need more time than this to complete your revisions, please reply to this message or contact the journal office at plosntds@plos.org. Please include the following items when submitting your revised manuscript:

We look forward to receiving your revised manuscript.

Kind regards,

Apichai Tuanyok, Ph.D.

Guest Editor

Stuart Blacksell

Section Editor

Shaden Kamhawi

co-Editor-in-Chief

Paul Brindley

co-Editor-in-Chief

**Journal Requirements:**

At this stage, the following Authors/Authors require contributions: Jinjuta Ngeyvijit, Chawakorn Payackpunth, Pattawee Saengmongkonpipat, Subencha Pinsai, Antenor Rodrigues, and Vorakamol Phoophiboon. Please ensure that the full contributions of each author are acknowledged in the "Add/Edit/Remove Authors" section of our submission form.

3) In the online submission form, you indicated that "The data that support the findings of this study are available on suitable requests to the corresponding authors". All PLOS journals now require all data underlying the findings described in their manuscript to be freely available to other researchers, either

- In a public repository

- Within the manuscript itself

- Uploaded as supplementary information.

**Reviewers' Comments:**

Reviewer's Responses to Questions

**Key Review Criteria Required for Acceptance?**

**Methods:**

-Are the objectives of the study clearly articulated with a clear testable hypothesis stated?

-Is the study design appropriate to address the stated objectives?

-Is the population clearly described and appropriate for the hypothesis being tested?

-Is the sample size sufficient to ensure adequate power to address the hypothesis being tested?

-Were correct statistical analysis used to support conclusions?

-Are there concerns about ethical or regulatory requirements being met?

Reviewer #1: 1. In a part of the introduction: the author should review the situation of melioidosis in Thailand and the study area, as all of the study population recruited in Thailand.

2. Line 37-38: Please clarify this sentence “Due to globalization, more cases of melioidosis have been reported outside of these areas(6-8).”. How does globalization affect the increase in melioidosis cases in other areas because it is not transmitted from humans to humans? Regarding the meaning of globalization, it is related to the high movement of people from one area to another and the easy connection among them.

3. Line 73: Please give a justification. Why is written informed consent not required for this study? In the methodology, the authors had to collect information (such as age, sex, weight, height, body mass index (BMI), comorbidities, history of excessive alcohol consumption, history of current smoking) for individuals who met the criteria, which included their data and risk factors. Collecting that information from patients in the study population requires consent. Is this correct?

Reviewer #2: The study seems well articulated with a good number of patients, considering the rarity of cases of melioidosis. Strong statistical analyses have been made to support the results of the research conducted. There is a minor concern about ethical issues as written informed consent have not been taken which the authors need to address.

**Results:**

-Does the analysis presented match the analysis plan?

-Are the results clearly and completely presented?

-Are the figures (Tables, Images) of sufficient quality for clarity?

Reviewer #1: 1. Line 151-152: the authors stated that Streptococcus viridans, Klebsiella pneumoniae, and Acinetobactor baumanii were concomitantly detected in sputum cultures. Is it possible that these patients with pulmonary melioidosis were admitted to the hospital and acquired those bacteria while they were there? In the discussion section, the author may include an issue of infection prevention control in the study hospital as a possible risk factor related to their respiratory outcome.

2. Line 160-161: result of univariate logistic regression analysis showed that ARDS Pulmonary melioidosis with ARDS was associated with body mass index (BMI) < 20 Kg/m2. The author should conduct a multivariate regression to explore any confounder factors. BMI can be associated with age and certain chronic diseases, but there isn't a causal pathway that links the effects of BMI to ARDS Pulmonary melioidosis.

3. Table 4: The authors used age as a category variable to compare between two groups. Is this correct? What is the reference group for age? If the author used age as a continuous variable, how were odd ratios calculated?

Reviewer #2: yes, correct analyses have been made with appropriate tables and figures.

**Conclusions:**

-Are the conclusions supported by the data presented?

-Are the limitations of analysis clearly described?

-Do the authors discuss how these data can be helpful to advance our understanding of the topic under study?

-Is public health relevance addressed?

Reviewer #1: 1. Line 207-208: Please provide any references that present the same results of lengths of stay in survival patients with non-ARDS in pulmonary melioidosis in hospital and non-ARDS cases in community acquired pneumonia. These references could provide some assumptions for future research on this topic.

2. Line 217-218: The author stated that “the study relied on retrospective data collection from a single center. Thus, our data cannot be used to infer causal relationships. These two sentences may not be relevant. Please consider that the single center can affect the limitation of generalizability related to making inferences about the target population.

Reviewer #2: Yes, the authors clearly stated their shortcomings and strength which seemed appropriate. the authors also mention, how the data can help in further addressing the risks associated with this rare disease.

**Editorial and Data Presentation Modifications?**

Reviewer #1: (No Response)

Reviewer #2: Minor revision required.

**Summary and General Comments:**

Reviewer #1: (No Response)

Reviewer #2: (No Response)

PLOS authors have the option to publish the peer review history of their article (what does this mean? ). If published, this will include your full peer review and any attached files.

**Do you want your identity to be public for this peer review?** For information about this choice, including consent withdrawal, please see our Privacy Policy .

Reviewer #1: **Yes: ** Soawapak Hinjoy

Reviewer #2: No

**Figure resubmission:**
---

## [Decision Letter · Decision Letter 1]

PNTD-D-24-01573R1Exploring Patient Characteristics and Respiratory Impacts of Pulmonary Melioidosis: A 5-Year Experience from Endemic Region of ThailandPLOS Neglected Tropical DiseasesDear Dr. Phoophiboon, Thank you for submitting your manuscript to PLOS Neglected Tropical Diseases. After careful consideration, we feel that it has merit but does not fully meet PLOS Neglected Tropical Diseases's publication criteria as it currently stands. Therefore, we invite you to submit a revised version of the manuscript that addresses the points raised during the review process. Please submit your revised manuscript within 30 days Jul 03 2025 11:59PM. If you will need more time than this to complete your revisions, please reply to this message or contact the journal office at plosntds@plos.org. Please include the following items when submitting your revised manuscript: * A rebuttal letter that responds to each point raised by the editor and reviewer(s). You should upload this letter as a separate file labeled 'Response to Reviewers '. This file does not need to include responses to any formatting updates and technical items listed in the 'Journal Requirements' section below. * A marked-up copy of your manuscript that highlights changes made to the original version. You should upload this as a separate file labeled 'Revised Manuscript with Track Changes '. * An unmarked version of your revised paper without tracked changes. You should upload this as a separate file labeled 'Manuscript '. If you would like to make changes to your financial disclosure, competing interests statement, or data availability statement, please make these updates within the submission form at the time of resubmission. Guidelines for resubmitting your figure files are available below the reviewer comments at the end of this letter. We look forward to receiving your revised manuscript. Kind regards, Apichai Tuanyok, Ph.D.Guest EditorPLOS Neglected Tropical Diseases Stuart BlacksellSection EditorPLOS Neglected Tropical Diseases

Shaden Kamhawi

co-Editor-in-Chief

Paul Brindley

co-Editor-in-Chief

**Reviewers' comments:** Reviewer's Responses to Questions

**Key Review Criteria Required for Acceptance?**

**Methods:**

-Are the objectives of the study clearly articulated with a clear testable hypothesis stated?

-Is the study design appropriate to address the stated objectives?

-Is the population clearly described and appropriate for the hypothesis being tested?

-Is the sample size sufficient to ensure adequate power to address the hypothesis being tested?

-Were correct statistical analysis used to support conclusions?

-Are there concerns about ethical or regulatory requirements being met?

Reviewer #3: Yes

**Results**

-Does the analysis presented match the analysis plan?

-Are the results clearly and completely presented?

-Are the figures (Tables, Images) of sufficient quality for clarity?

Reviewer #3: Yes

**Conclusions:**

-Are the conclusions supported by the data presented?

-Are the limitations of analysis clearly described?

-Do the authors discuss how these data can be helpful to advance our understanding of the topic under study?

-Is public health relevance addressed?

Reviewer #3: Yes

**Editorial and Data Presentation Modifications?**

Reviewer #3: Minor revision

**Summary and General Comments:**

Reviewer #3: Ref: PNTD-D-24-01573R1

Review Comments:

The revised manuscript has been improved, and the authors have addressed all previous reviewer concerns.

This study presents valuable clinical data from an endemic region in Thailand, contributing to the understanding of respiratory manifestations of pulmonary melioidosis and factors associated with ARDS development.

One minor issue remains:

The manuscript includes numerous technical terms and abbreviations (e.g., HFNC, MV in line 161) that are not always accompanied by their full definitions upon first use. For clarity and accessibility, especially for non-specialist readers, I recommend using full terms or ensuring that all abbreviations are fully defined when first introduced in both the abstract and main text. 

PLOS authors have the option to publish the peer review history of their article (what does this mean? ). If published, this will include your full peer review and any attached files.

**Do you want your identity to be public for this peer review?** For information about this choice, including consent withdrawal, please see our Privacy Policy .

Reviewer #3: **Yes: ** Narisara Chantratita

---

## [Editor Report · Decision Letter 2]

Dear Dr. Phoophiboon,

We are pleased to inform you that your manuscript 'Exploring Patient Characteristics and Respiratory Impacts of Pulmonary Melioidosis: A 5-Year Experience from Endemic Region of Thailand' has been provisionally accepted for publication in PLOS Neglected Tropical Diseases.

Best regards,

Apichai Tuanyok, Ph.D.

Guest Editor

Stuart Blacksell

Section Editor

Shaden Kamhawi

co-Editor-in-Chief

Paul Brindley

co-Editor-in-Chief

---

## [Editor Report · Acceptance letter]

Dear Dr. Phoophiboon,

We are delighted to inform you that your manuscript, "Exploring Patient Characteristics and Respiratory Impacts of Pulmonary Melioidosis: A 5-Year Experience from Endemic Region of Thailand," has been formally accepted for publication in PLOS Neglected Tropical Diseases.

Best regards,

Shaden Kamhawi

co-Editor-in-Chief

Paul Brindley

co-Editor-in-Chief
